# Generation of Geodesics with Actor-Critic Reinforcement Learning to Predict Midpoints

## Abstract

Various tasks in the real world, such as path planning, can be reduced to the generation of geodesics on manifolds. For reinforcement learning to generate geodesics sequentially, we need to define rewards appropriately. To generate geodesics without any adjustment of rewards, we propose to use a modified version of sub-goal trees, called *midpoint trees*. While sub-goal trees consist of arbitrary intermediate points, midpoint trees consist of midpoints. In addition, we propose an actor-critic method to learn to predict midpoints and theoretically prove that, under mild assumptions, when the learning converges at the limit of infinite tree depth, the resulting policy generates exact midpoints. We show experimentally that our proposed method outperforms existing methods in a certain path planning task.

## 1 Introduction

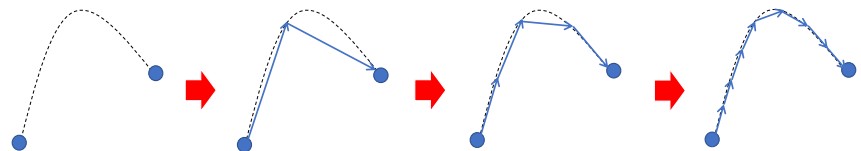

Figure 1: Midpoint tree generation of a geodesic (dotted curve)

On Riemannian manifolds, or more generally on Finsler manifolds, geodesics are curves connecting points locally with minimum lengths. Various tasks in the real world can be reduced to the generation of geodesics on manifolds. An example is time-optimal path planning on sloping ground (Matsumoto, 1989). A framework for robot motion planning by formulating it using Riemannian geometry has also been studied (Ratliff et al., 2015). Typically, metrics are only known infinitesimally (i.e., a form of a Riemannian or Finsler metric) and their distance functions are not known beforehand. To compute geodesics by optimizing curves or solving differential equations is generally computationally costly and needs an explicit form of the metric or, at least, its differentials. When we want to find geodesics for various pairs of endpoints in a fixed manifold, it is more efficient to use a policy that has learned to generate geodesics for arbitrary pairs.

Goal-conditioned reinforcement learning (Schaul et al., 2015) to generate waypoints of geodesics sequentially from start to goal points suffers from the sparseness of reward if an agent gets rewards only when it reaches a goal. To overcome this, a typical strategy is to give the agent rewards when it gets near its goal. However, to define the values of rewards, we must know some appropriate approximation of the distance function between two points beforehand. When metrics of manifolds are complex, it may be difficult to find an appropriate approximation of the distance function. Furthermore, when trying to generate numerous waypoints, the long horizon makes learning difficult (Wang et al., 2020).

To overcome these difficulties, we propose a framework called the *midpoint tree*, a modification of the sub-goal tree framework proposed by Jurgenson et al. (2020). In the sub-goal tree framework, a policy learns to predict an intermediate point between two given points, instead of the next point as in the sequential framework. Paths between the start and goal points are generated by recursively applying this prediction to adjacent pairs of previously generated waypoints. In our midpoint tree

framework, as shown in Fig. 1, instead of an arbitrary intermediate points, the policy predicts a midpoint, which is an intermediate point whose distances from two given points are equal. This modification enables appropriate generation when metrics are only locally known. Compared to sequential generation, these recurrent generation methods also have the advantage that they can be naturally parallelized.

The original learning method used to predict sub-goals using policy gradient in Jurgenson et al. (2020) has poor sample efficiency when trees are deep. To improve upon this, we also propose an actor-critic learning method for predicting midpoints, which is similar to the actor-critic method (Konda & Tsitsiklis, 1999) for conventional reinforcement learning. We prove theoretically that, under mild assumptions, if the training converges in the limit of infinite recursion depth, the resulting predictions will coincide with the true midpoints. This result does not hold for generation by arbitrary intermediate points.

Experimentally, we compared our proposed method with sequential generation with goal-conditioned reinforcement learning and midpoint tree generation trained by a policy gradient method without a critic in two practical path planning problems. The first considers a sloped ground and the second consider non-holonomic vehicles. In the latter task, our method clearly outperformed baseline methods. In addition, we ran our method and baselines on a motion planning task for a 7 DoF robotic arm in an obstacle environment to confirm the effectiveness of our method for collision-free motion planning of robots.

## 2 RELATED WORKS

### 2.1 PATH PLANNING WITH REINFORCEMENT LEARNING

One of the most popular approaches for path planning via reinforcement learning is to use a Q-table (Haghzad Klidbary et al., 2017; Low et al., 2022). However, such an approach depends on the finiteness of state spaces and computational costs grow with respect to the sizes of these spaces.

Several studies have been conducted on path planning in continuous space using deep reinforcement learning (Zhou et al., 2019; Wang et al., 2019; Kulathunga, 2022). In these works, methods depend on custom rewards.

### 2.2 GOAL-CONDITIONED REINFORCEMENT LEARNING AND SUB-GOALS

Goal-conditioned reinforcement learning (Kaelbling, 1993; Schaul et al., 2015) trains a universal policy for various goals. It learns a value function that inputs both current and goal states. Kaelbling (1993) and Dhiman et al. (2018) pointed out that goal-conditioned value functions are related to the Floyd-Warshall algorithm for the all pairs shortest path problem (Floyd, 1962), as this function can be updated by finding intermediate states. They proposed methods that use brute force to search for intermediate states, which depend on the finiteness of state spaces. The idea of using sub-goals for reinforcement learning is suggested by Sutton et al. (1999) as options, and Jurgenson et al. (2020) linked this notion to the aforementioned intermediate states. Wang et al. (2023) drew attention to the quasi-metric structure of goal-conditioned value functions and suggested using quasi-metric learning (Wang & Isola, 2022) to learn these functions.

The idea of generating paths by predicting sub-goals recursively has been proposed in three papers with different problem settings and methods. The problem setting for goal-conditioned hierarchical predictors by Pertsch et al. (2020) differs from ours because they use an approximate distance function learned from given training data, where no training data are given in our setting. While the divide-and-conquer Monte Carlo tree search by Parascandolo et al. (2020) is similar to our method because they train both the policy prior and the approximate value function, which correspond to the actor and critic in our method, their algorithm depends on the finiteness of the state spaces.

The problem setting in Jurgenson et al. (2020) is most similar to ours but different. In their setting, the costs of direct edges between two points are given, which are upper bounds for distances. In our setting, we can only approximate distances between two points when they are close. Therefore, we must find waypoints such that the adjacent points are close. This is one of the reasons why we use midpoint trees instead of sub-goal trees. In addition, because they focus on high-level planning,

their trees are not as deep as ours. This is one of the reasons why we propose an actor-critic method while they use a policy gradient method without a critic.

# 3 PRELIMINARY

## 3.1 QUASI-METRIC SPACE

We follow the notation in Kim (1968). Let $X$ be a space. A *pseudo-quasi-metric* on $X$ is a function $d : X \times X \to [0, \infty)$ such that $d(x, x) = 0$ and $d(x, z) \le d(x, y) + d(y, z)$ for any $x, y, z \in X$. A topology on $X$ is induced by $d$, which has the collection of all open balls $\{y \in X | d(x, y) < r\}$ as a base. A pseudo-quasi-metric $d$ is called a *quasi-metric* if $d(x, y) > 0$ for any $x, y \in X$ with $x \ne y$.

A pseudo-quasi-metric $d$ is called *weakly symmetric* if $d(x, y_i) \to 0$ indicates $d(y_i, x) \to 0$, for any $x \in X$ and any sequence $(y_i)_i$ (Arutyunov et al., 2017). When $d$ is weakly symmetric, $d$ is continuous as a function with respect to the topology induced by $d$.

For two points $x, y \in X$, a point $z \in X$ is called a *midpoint* between $x$ and $y$ if $d(x, z) = d(z, y) = d(x, y)/2$. $(X, d)$ is said to have the *midpoint property* if there exists at least one midpoint for every pair of points. $(X, d)$ is said to have the *continuous midpoint property* if there exists a continuous map $m : X \times X \to X$ such that $m(x, y)$ is a midpoint between $x$ and $y$ for any $x, y \in X$ (Horvath, 2009).

## 3.2 FINSLER GEOMETRY

An important family of quasi-metric spaces is the Finsler manifolds. A *Finsler manifold* is a differential manifold $M$ equipped with a function $F : TM \to [0, \infty)$, where $TM = \bigcup_{x \in M} T_x M$ is the tangent bundle of $M$, satisfying the following conditions (Bao et al., 2000).

1. $F$ is smooth on $TM \setminus 0$.
2. $F(x, \lambda v) = \lambda F(x, v)$ for all $\lambda > 0$ and $(x, v) \in TM$.
3. The Hessian matrix of $F(x, -)^2$ is positive definite at every point of $TM_x \setminus 0$ for all $x \in M$.

Let $\gamma : [0, 1] \to M$ be a piecewise smooth curve on $M$. The length of $\gamma$ is defined as

$$L(\gamma) := \int_0^1 F\left(\gamma(t), \frac{d\gamma}{dt}(t)\right) dt. \tag{1}$$

For two points $x, y \in M$, the distance $d(x, y)$ is defined as

$$d(x, y) := \inf \{L(\gamma) | \gamma : [0, 1] \to M, \gamma(0) = x, \gamma(1) = y\}. \tag{2}$$

Then, $d$ is a weakly symmetric quasi-metric (Bao et al., 2000). A curve $\gamma : [0, 1] \to M$ is called a *minimizing geodesic* if $L(\gamma) = d(\gamma(0), \gamma(1))$.

The following fact is known (Bao et al., 2000; Amici & Casciaro, 2010). For any point of $M$, there exists its neighborhood $U \subseteq M$ such that any two points $p, q \in U$ can be connected by a minimizing geodesic inside $U$ uniquely. Note that $(U, d)$ has the continuous midpoint property.

# 4 LEARNING METHOD

In this section, we describe our proposed learning method for predicting midpoints.

## 4.1 SETTING

Let $(X, d)$ be a pseudo-quasi-metric space. We assume that $d$ is not known, but the metric is known infinitesimally, i.e., we have a continuous function $C : X \times X \to \mathbb{R}$ such that $C(x, y)/d(x, y) \to 1$ when $d(x, y) \to 0$ or $C(x, y) \to 0$. Formally, we assume the following conditions.

1. For $x \in X$ and a series $(y_i)_i$ on $X$, $d(x, y_i) \to 0$ if $C(x, y_i) \to 0$.

2. For $x \in X$ and $\varepsilon > 0$, there exists $\delta > 0$ such that for any $y, z \in X$,

$$d(x,y) < \delta \wedge d(x,z) < \delta \implies (1-\varepsilon)d(y,z) \leq C(y,z) \leq (1+\varepsilon)d(y,z). \quad (3)$$

We want to find a function $m : X \times X \to X$ such that $d(x, m(x,y)) = d(m(x,y), y) = d(x,y)/2$ for any $x, y \in X$.

We mainly consider cases where $(X, d)$ is a Finsler manifold $(M, F)$ with a global coordinate system (diffeomorphism to a subset) $f : M \hookrightarrow \mathbb{R}^d$. We can define $C$ as

$$C(x,y) := F\left(x, df_x^{-1}(f(y) - f(x))\right), \quad (4)$$

where $df_x : T_x M \to T_{f(x)} \mathbb{R}^d = \mathbb{R}^d$ is the differential of $f$ at $x$.

**Proposition 1.** *$C$ satisfies the aforementioned conditions.*

See Appendix A.1 for the proof.

**Remark 1.** When $M$ is compact, once a wanted function $m : M \times M \to M$ is found, we can construct minimizing geodesics for all pairs of points. Let $A := \{N/2^n | n \geq 0, 0 \leq N \leq 2^n\}$. For any $x, y \in M$, by applying $m$ recursively, we can construct $\gamma : A \to M$ such that $d(x, \gamma(a)) = ad(x,y)$ and $d(\gamma(a), y) = (1-a)d(x,y)$ for any $a \in A$. For any $r \in [0,1]$, we can take a non-decreasing sequence $a_1, a_2, \ldots \in A$ such that $\lim_i a_i = r$. Then, since $\gamma(a_i)$ is a forward Cauchy sequence with respect to $d$, it converges (Bao et al., 2000). Therefore, we can extend the domain of $\gamma$ to $[0,1]$.

For further remarks on this setting, see Appendix C.

## 4.2 ALGORITHM

We simultaneously train two networks called the *actor* and the *critic*. The actor $\pi_\theta$ predicts midpoints between two given points and the critic $V_\phi$ predicts distances of two given points, where $\theta$ and $\phi$ are their parameters. The critic learns to predict distances from lengths of sequences generated by the actor, and the actor learns to predict midpoints from predictions by the critic.

The network for actor $\pi_\theta$ has a form for which the reparametrization trick (Haarnoja et al., 2018) can be used, i.e., a sample is drawn by computing a deterministic function of input, parameters $\theta$, and independent noise. Therefore, we abuse symbols and a sampled prediction from $\pi_\theta(\cdot|s, g)$ is simply denoted by $\pi_\theta(s, g)$ even if it is not deterministic.

Alg. 1 shows the pseudocode of our methods. We define the loss $L_{\text{critic}}$ for a critic $V$ and $s, g \in X$ with an estimated distance $c$ as

$$L_{\text{critic}}(V, s, g, c) := (V(s,g) - c)^2. \quad (5)$$

The loss $L_{\text{actor}}$ for an actor $\pi$ with a critic $V$ and $s, g \in X$ is defined as

$$L_{\text{actor}}(\pi, V, s, g) := V(s, \pi(s,g))^2 + V(\pi(s,g), g)^2. \quad (6)$$

The expression is intended to make $\pi(s, g)$ a midpoint between $s$ and $g$. Note that $d(x, z)^2 + d(z, y)^2$ takes the minimum value when $z$ is a midpoint between $x$ and $y$ since

$$d(x,z)^2 + d(z,y)^2 = \frac{1}{2}\left((d(x,z) + d(z,y))^2 + (d(x,z) - d(z,y))^2\right) \geq \frac{1}{2}d(x,y)^2. \quad (7)$$

The data for training is collected using the actor $\pi_\theta$ with the current parameters. We sample two points from $X$ and generate a sequence of points by repeatedly inserting points between adjacent points with $\pi_\theta$. Adjacent pairs of points at each iteration are collected as data. The depth of iteration is not necessarily constant and can be scheduled. The estimated distances of collected pairs are simply calculated as sums of values of $C$ for adjacent pairs in the final sequence between them. In other words, we use a Monte Carlo method.

After collecting data, we split them to mini-batches and repeat $N_{\text{epochs}}$ times, update the parameters of the actor and critic in accordance with gradients of the aforementioned losses for each mini-batch, using an optimization algorithm. We repeat this process of data collection and optimization enough times.

**Remark 2.** Instead of the Monte Carlo method, we may use TD($\lambda$) (Sutton & Barto, 2018) for $0 \leq \lambda \leq 1$ as $c_{D,j} := C(p_{D,j}, p_{D,j+1})$ and, for $i = D-1, \ldots, 0$,

$$c_{i,j} := (1-\lambda)(V_\phi(p_{i,j}, p_{i+1,2j+1}) + V_\phi(p_{i+1,2j+1}, p_{i,j+1})) + \lambda(c_{i+1,2j} + c_{i+1,2j+1}). \quad (8)$$

---

**Algorithm 1** Actor-Critic Midpoint Learning

---

1: Initialize $\theta, \phi$
2: **while** learning is not done **do**
3:     $data \leftarrow \emptyset$
4:     **while** $data$ is not enough **do**
5:         $data \leftarrow data \cup \text{COLLECTDATA}(\pi_\theta)$
6:     **end while**
7:     Split $data$ to $batches$
8:     **for** $epoch = 1, \ldots, N_{\text{epochs}}$ **do**
9:         **for all** $b \in batches$ **do**
10:             Update $\phi$ in accordance with $\nabla_\phi \sum_{(s,g,c)\in b} L_{\text{critic}}(V_\phi, s, g, c)$
11:             Update $\theta$ in accordance with $\nabla_\theta \sum_{(s,g,c)\in b} L_{\text{actor}}(\pi_\theta, V_\phi, s, g)$
12:         **end for**
13:     **end for**
14: **end while**
15:
16: **procedure** COLLECTDATA$(\pi)$
17:     Sample two points $p_{0,0}, p_{0,1}$
18:     Decide the depth $D$
19:     **for** $i = 1, \ldots, D$ **do**
20:         $p_{i,0}, \ldots, p_{i,2^i} \leftarrow p_{i-1,0}, \pi(p_{i-1,0}, p_{i-1,1}), p_{i-1,1}, \pi(p_{i-1,1}, p_{i-1,2}), \ldots, p_{i-1,2^{i-1}}$
21:     **end for**
22:     $c_0, \ldots, c_{2^D-1} \leftarrow C(p_{D,0}, p_{D,1}), C(p_{D,1}, p_{D,2}), \ldots, C(p_{D,2^D-1}, p_{D,2^D})$
23:     $data \leftarrow \{(p_{D,0}, p_{D,0}, 0), (p_{D,1}, p_{D,1}, 0), \ldots, (p_{D,2^D}, p_{D,2^D}, 0)\}$
24:     **for** $i = D, \ldots, 0$ and $j = 0, \ldots, 2^i - 1$ **do**
25:         $c_{i,j} \leftarrow c_{2^{D-i}j} + c_{2^{D-i}j+1} + \cdots + c_{2^{D-i}(j+1)-1}$
26:         $data \leftarrow data \cup \{(p_{i,j}, p_{i,j+1}, c_{i,j})\}$
27:     **end for**
28:     **return** $data$
29: **end procedure**

---

## 4.3 UNIQUENESS OF SOLUTION

Obtaining our desired outcome after the convergence of our learning method is a non-trivial achievement. Strictly speaking, our method can not converge completely for finite depth $D$. Instead, we assume that $D$ is sufficiently large and consider the limit for $D \to \infty$. Instead of a single actor and critic, we consider the optimal $\pi_i^*$ and $V_i^*$ dependent on the recursion depth $i$ (how many times the recursion follows) as in Jurgenson et al. (2020). $V_i^*$ represents the sum of the values of $C$ at two consecutive points for the sequence obtained by applying policies $\pi_{i-1}^*, \ldots, \pi_0^*$, while $\pi_i^*$ is a policy to minimize $V_i^*(x, \pi_i^*(x,y))^2 + V_i^*(\pi_i^*(x,y), y)^2$. In other words,

$$V_0^*(x, y) = C(x, y), \tag{9}$$

$$\pi_i^*(x, y) \in \arg\min_z \left( V_i^*(x, z)^2 + V_i^*(z, y)^2 \right), \tag{10}$$

$$V_{i+1}^*(x, y) = V_i^*(x, \pi_i^*(x, y)) + V_i^*(\pi_i^*(x, y), y). \tag{11}$$

Under mild assumptions, the limits of $(\pi_i^*)_i$ and $(V_i^*)_i$ coincide with our desired outcome.

**Proposition 2.** *We assume that $(X, d)$ is a compact weakly symmetric pseudo-quasi-metric space with the midpoint property and there exists a series of functions $\pi_i^* : X \times X \to X$ and $V_i^* : X \times X \to \mathbb{R}$ satisfying (9), (10), and (11). We also assume that $(V_i^*)_i$ are equicontinuous, i.e., for any $x, y \in X$ and $\varepsilon > 0$, there exists $\delta > 0$ such that for any $x', y' \in X$ and $i$,*

$$d(x, x') < \delta \wedge d(y, y') < \delta \implies |V_i^*(x, y) - V_i^*(x', y')| < \varepsilon. \tag{12}$$

*Then, if $\pi^* : X \times X \to X$ and $V^* : X \times X \to \mathbb{R}$ are continuous functions such that $d(\pi^*(x, y), \pi_i^*(x, y)) \to 0$ and $V_i^*(x, y) \to V^*(x, y)$ when $i \to \infty$ for any $x, y \in X$, $V^* = d$ and $\pi^*(x, y)$ is a midpoint between $x$ and $y$ for any $x, y \in X$.*

For the proof, see Appendix A.2.

**Remark 3.** If (10) is replaced with

$$\pi_i^*(x, y) \in \arg\min_z \left( V_i^*(x, z) + V_i^*(z, y) \right), \tag{13}$$

the conclusion does not follow. Let $f : [0, \infty) \to [0, \infty)$ be a non-decreasing subadditive function such that $\lim_{h \to +0} f(h)/h = 1$ (For example, $f(h) := 2\sqrt{h + 1} - 2$) and let $C := f \circ d$. We consider the case where $V_i^* = V^* = C$ and $\pi_i^*(x, y) = \pi^*(x, y) = x$. Then, (13) and the conditions of Proposition 2 except (10) hold. However, $V^* \neq d$ generally.

Note that $V^* \leq d$ follows even from these conditions. This fact supports our following insights. If upper bounds of distances are given, the approach to predict arbitrary intermediate points can work as in Jurgenson et al. (2020). However, if distances can be approximated for only two close points, it is necessary to avoid generating points near endpoints.

## 5 EXPERIMENTS

We compared our method, which generates geodesics by policies trained by the method in the previous section, with baseline methods in two path planning tasks. In addition, we also ran the proposed and baseline methods in a robot motion planning task.

### 5.1 TASKS AND EVALUATION METHOD

We compared methods by the success rate of solving the following task. A Finsler manifold $(M, F)$ with a global coordinate system $f : M \hookrightarrow \mathbb{R}^d$ is given as an environment. The number $n$ of segments to approximate curves and a threshold for proximity $\varepsilon > 0$ are also fixed. For our method, $n$ must be a power of two: $n = 2^{D_{\max}}$, where $D_{\max}$ is the depth of midpoint trees. When two points $s, g \in M$ are given, we want to generate a sequence $s = p_0, p_1, \ldots, p_n = g$ of points such that all values of $C$ (4) for two consecutive points are not greater than $\varepsilon$. If the points generated by a method satisfy this condition, it is considered successful in solving this task, otherwise, it is considered to have failed.[1]

For each environment, we generated 100 pairs of points randomly before the experiment. During training, we evaluated models regularly by solving the aforementioned tasks for generated pairs and recorded the success rate. For the environments in § 5.3.1 and § 5.3.2, we ran each method 10 times with different random seeds. For the environment in § 5.3.3, only one run was performed for each method, for reasons of computational cost.

### 5.2 BASELINE METHODS

The baseline methods are as follows:

- **Sequential Reinforcement Learning (Seq)**: We formulated a sequential generation of waypoints as a conventional goal-conditioned reinforcement learning environment. The agent moves to the goal step by step in $M$. If the agent is at $p$, it can move to $q$ such that $C(p, q) = \varepsilon$. The reward $R$ is defined as

$$R := F\left(g, df_g^{-1}(f(g) - f(p))\right) - F\left(g, df_g^{-1}(f(g) - f(q))\right) - \varepsilon, \tag{14}$$

  where $g$ is the goal. The reason that we do not define reward by using $C$ is that $C(p, g)$ does not necessarily decrease when $p$ gets closer to $g$. The discount factor is set to one. An episode ends and is considered as success when the agent reaches a point $p$ close enough to the goal as $C(p, g) < \varepsilon$. When the episode duration reaches $n$ steps without satisfying the aforementioned condition, the episode ends and considered a failure.

  We used Proximal Policy Optimization (PPO) (Schulman et al., 2017) to solve reinforcement learning problems with this formulation.

---

[1]The reason we do not use the lengths of paths for evaluation is that, since the metric can only be calculated locally, lengths cannot be calculated unless the success condition is satisfied. Furthermore, this evaluation scheme allows for fair comparisons with the baseline method with sequential generation.

- **Policy Gradient (PG)**: We modified the method in Jurgenson et al. (2020) to predict midpoints by changing values to optimize while training. For each $D = 1, \ldots, D_{\max}$, a stochastic policy $\pi_D$ is trained to predict midpoints on the depth $D$. To generate of waypoints, we apply policies in descending order of index. We train policies in ascending order of index by policy gradient.

  Let $\rho(\pi_1, \ldots, \pi_D)$ be the distribution of $\tau := (p_0, \ldots, p_{2^D})$, where $p_0$ and $p_{2^D}$ is sampled from the predefined distribution on $M$ and $p_{2^{i-1}(2j+1)}$ is sampled from the distribution $\pi_i \left( \cdot \middle| p_{2^i j}, p_{2^i(j+1)} \right)$. Let $\theta_D$ denote the parameters of $\pi_D$. Instead of minimizing the expected value of $\sum_{i=0}^{2^D-1} C(p_i, p_{i+1})$ as in the original method, we train $\pi_D$ to minimize the expected value of

$$c_\tau := \left( \sum_{i=0}^{2^{D-1}-1} C(p_i, p_{i+1}) \right)^2 + \left( \sum_{i=2^{D-1}}^{2^D-1} C(p_i, p_{i+1}) \right)^2, \tag{15}$$

  using

$$\nabla_{\theta_D} \mathbb{E}_{\tau \sim \rho(\pi_1, \ldots, \pi_D)} [c_\tau] = \mathbb{E}_{\tau \sim \rho(\pi_1, \ldots, \pi_d)} \left[ (c_\tau - b(p_0, p_{2^D})) \nabla_{\theta_D} \log \pi_d \left( p_{2^{D-1}} | p_0, p_{2^D} \right) \right], \tag{16}$$

  where $b$ is a baseline function.

  When the model is evaluated while training, if the current trained policy is $\pi_D$ ($1 \le D \le D_{\max}$), the evaluation is done with depth $D$ ($2^D$ segments), while other methods are always evaluated with $n = 2^{D_{\max}}$ segments.

In addition to our proposed method (**Ours**) described in §4.2, we ran its following variants.

- **Intermediate Point (Inter)**: Instead of (6), we use the following actor loss:

$$L_{\text{actor}}(\pi, Q, s, g) := V(s, \pi(s, g)) + V(\pi(s, g), g). \tag{17}$$

  This means that $\pi$ learns to predict intermediate points that are not necessary midpoints.

- **2:1 Point (2:1)**: Instead of (6), we use the following actor loss:

$$L_{\text{actor}}(\pi, Q, s, g) := V(s, \pi(s, g))^2 + 2V(\pi(s, g), g)^2. \tag{18}$$

  This means that $\pi$ learns to predict $2:1$ points instead of midpoints since

$$d(x, z)^2 + 2d(z, y)^2 = \frac{1}{3} \left( 2(d(x, z) + d(z, y))^2 + (d(x, z) - 2d(z, y))^2 \right). \tag{19}$$

For all methods, timesteps are measured by counting the evaluation of $C$ while training. In other words, for sequential RL, timesteps have the conventional meaning. For other methods, the evaluation of a generated path with depth $D$ is counted as $2^D$ timesteps.

## 5.3 ENVIRONMENTS

We experimented in the following three environments.

### 5.3.1 MATSUMOTO METRIC

*Matsumoto metric* is an asymmetric Finsler metric considering times to move in inclined planes, introduced by Matsumoto (1989). Let $M \subseteq \mathbb{R}^2$ be a region on the plane with standard coordinates $x, y$ and let $h : M \to \mathbb{R}$ be a differentiable function, which means heights of the field. The Matsumoto metric $F : TM \to [0, \infty)$ is defined as follows:

$$F(p, v_x dx + v_y dy) := \frac{\alpha^2}{\alpha - \beta} \tag{20}$$

where,

$$\beta := v_x \frac{\partial h}{\partial x}(p) + v_y \frac{\partial h}{\partial y}(p), \ \alpha := \sqrt{v_x^2 + v_y^2 + \beta^2}. \tag{21}$$

We take a unit disk as $M$ and $h(p) := -\|p\|^2$. Intuitively, we consider a round field with a mountain at the center. We set the number of segments $n = 64$ (depth $D_{\max} = 6$) and $\varepsilon = 0.1$. We trained all methods with approximately $2 \times 10^7$ timesteps.

**Remark 4.** This environment seems not have the continuous midpoint property since minimizing geodesics can switch between counterclockwise and clockwise routes. However, our method can work as shown in § 5.4.

### 5.3.2 UNIDIRECTIONAL CAR-LIKE CONSTRAINTS

Inspired by a cost function for trajectories of car-like non-holonomic vehicles in Rösmann et al. (2017), we define a quasi-metric for unidirectional car-like vehicles.

Let $M := U \times [-\pi, \pi] \subseteq \mathbb{R}^3$ be a configuration space for car-like vehicles, where $U \subseteq \mathbb{R}^2$ is a region with a standard coordinate system $x, y$. Let $\theta$ denotes the third coordinate representing angles. We define $F : TM \to [0, \infty)$ as follows:

$$F((p, \theta), v_x dx + v_y dy + v_\theta d\theta) := \sqrt{v_x^2 + v_y^2 + c_p(h^2 + \xi^2)}, \tag{22}$$

where

$$h := -v_x \sin(\theta) + v_y \cos(\theta), \tag{23}$$

$$\xi := \max \{r_{\min}|v_\theta| - v_x \cos(\theta) - v_y \sin(\theta), 0\}, \tag{24}$$

$c_p$ is a penalty coefficient, and $r_{\min}$ is a lower bound of radius of curvature. The term $h$ penalizes moving sideways, and the term $\xi$ penalizes moving backward and sharp turns. Note that, since we do not identify $U \times \{-\pi\}$ and $U \times \{\pi\}$, the path traversing the angle corresponding $-\pi$ and $\pi$ cannot be considered.

Strictly speaking, this metric is not a Finsler metric, since $F$ is not smooth on the entire $TM \setminus 0$. See Appendix D.

In this experiment, we take a unit disk as $U$ and set $c_p := 100$, $r_{\min} := 0.5$. We set the number of segments $n = 256$ (depth $D_{\max} = 8$) and $\varepsilon = 0.1$. We trained all methods with approximately $8 \times 10^7$ timesteps.

### 5.3.3 7-DOF ROBOTIC ARM MOTION PLANNING WITH A COLUMNAR OBSTACLE

This environment is defined for motion planning in an environment with a columnar obstacle for Franka Panda robotic arm, which has 7 degrees of freedom (DoF). The space is the configuration space of the robot. The metric is warped around the obstacle for avoiding it (Mainprice et al., 2016). For the detail, see Appendix E.1.

We set the number of segments $n = 32$ (depth $D_{\max} = 5$). We trained all methods with approximately $2 \times 10^7$ timesteps.

### 5.4 RESULTS AND DISCUSSION

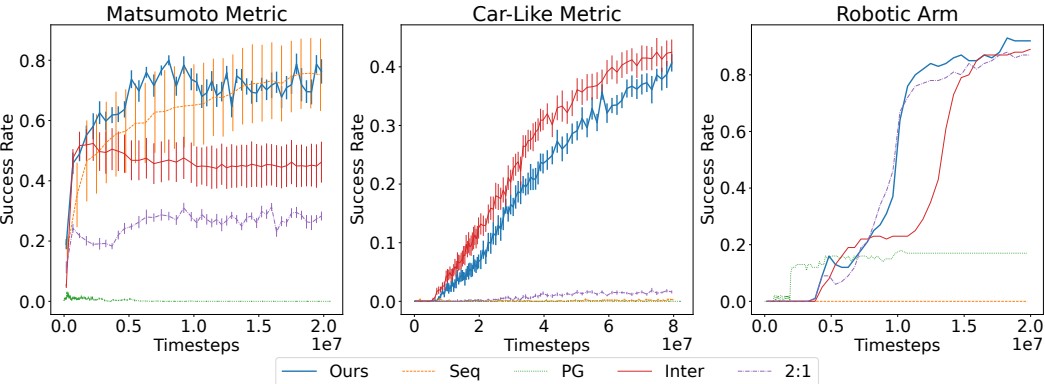

Figure 2: Plots of success rate

Fig. 2 shows the learning curves of success rate for all methods. Error bars represent standard errors of means. In the Matsumoto metric environment, the success rates for **Ours** and **Seq** approximately

75%. Standard errors are smaller for **Ours**, which means it is more stable for random seeds. This may be due to differences in the lengths of the horizons. The success rate for **Inter** decreases slightly from a certain point, which may be occurred by convergence to biased generation as in Remark 3. The success rate for **2:1** reaches only approximately 30%. **PG** was unable to solve most tasks. In the car-like metric environment, for **Ours** and **Inter**, the success rates reach approximately 40% and are likely to increase more if they keep learning. The success rates for other methods are low. Note that, while changing the reward definition (14) could possibly make learning successful in the sequential RL method, our method was successful without adjusting rewards. In the robotics arm environment, the success rates for our method and its variants exceed 80%, while that for **PG** is about 20% and that for **Seq** is zero. In the car-like metric environment and the robotic arm environment, **Inter** reaches about the same success rate as **Ours**, probably because points that minimize sums of approximated distances happen to be close to midpoints in these environments. Note that this is not always the case, as shown in Remark 3.

Fig. 3 shows examples of paths in the Matsumoto environment generated by the trained policy for each method. Every eight points are marked. Circles represent contours. The 'Truth' curve represents the ground truth of the minimizing geodesic with points dividing eight equal-length parts. In this example, all method except **PG** were able to generate curves close to the ground truth. While **Ours** generated waypoints near to the points dividing equally in the true curve, both **Inter** and **2:1** generation produced non-uniform waypoints. **PG** even failed to generate a smooth curve. This may be simply due to insufficient training. Note that **PG** gets only one training data per path generation, while our method gets several data on the scale of two to the power of the depth per path generation.

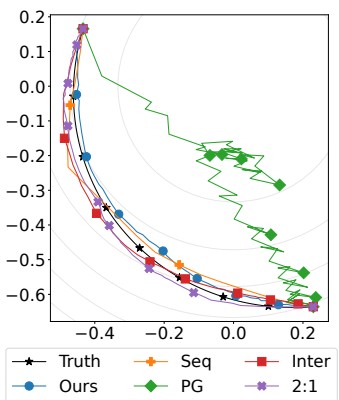

Figure 3: Generated paths

For an example of generated motion in the robotic arm environment, see Appendix E.2.

## 6 CONCLUSION AND FUTURE WORK

In this paper, we proposed a framework, called midpoint tree, to generate geodesics by recursively predicting midpoints. We also propose an actor-critic learning method for learning to predict midpoints and theoretically prove its soundness. Experimentally, we show that our method can solve a path planning task that existing reinforce learning methods fail to solve.

In this paper, we tried only a straightforward actor-critic learning algorithm, while algorithms for actor-critic reinforcement learning have been intensively researched. The investigation into more efficient algorithms is left for future work. In particular, while our algorithm is an on-policy one, off-policy algorithms (Degris et al., 2012) may be useful in our framework. While the architectures we used for both actors and critics were also simple, the quasi-metric learning method (Wang & Isola, 2022; Wang et al., 2023) may be useful for critics in our method.

One limitation of our method is that the policy has to be learned for each environment. By modifying our method so that the actor and critic input information on environments, it may be possible to learn a policy applicable to different environments. While we assume the continuity of policy function to prove the theoretical soundness of our method, the continuous midpoint property may be satisfied only locally. Further research is needed on the actual necessity of this assumption. Even if our method only work well locally, we could consider dividing manifolds, training policies for each region, and then connecting locally generated geodesics.

One possible extension of this work is the search for applications. While our experiment is focused on path planning tasks, Finsler geometry appears in several fields. For example, Finsler geometry is used in describing various physical systems (Pfeifer, 2019). Thus, our method may be useful for physical simulations. The Wasserstein distance, which is a metric for probabilistic distributions and used in several fields of computer science, is a Finsler metric (Aguih, 2012). Our method may have several applications where Wasserstein distance appears. Our method may be used for image morphing since it can be formulated as geodesics (Michelis & Becker, 2021; Effland et al., 2021).

## 7 REPRODUCIBILITY STATEMENT

The complete proofs of propositions we claim in this paper are provided in Appendix A. The implementation details and hyperparameter settings for both the proposed and baseline methods in our experiments are provided in Appendix B. In addition, the scripts for our experiments, including the designation of random seeds, and a dockerfile to reproduce the experimental environment are submitted as supplementary materials.

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

## A   PROOFS

### A.1   PROOF OF PROPOSITION 1

To prove that the first condition is satisfied, it is sufficient to show $f(y_i) \to f(x)$ when $C(x, y_i) \to 0$ because the topology induced by $d$ coincides with the topology of the underlying manifold (Bao et al., 2000). We can take the minimum value $c > 0$ of $F\left(x, f_x^{-1}(\mathbf{v})\right)$ for $\mathbf{v} \in S^{d-1} := \{v \in \mathbb{R}^d | \|v\| = 1\}$ since $S^{d-1}$ is compact. Then $C(x, y) \geq c\|f(y) - f(x)\|$ for any $y \in X$, from which what we want to show follows.

We prove that the second condition is satisfied. We fix $x \in X$. For a curve $\gamma : [0, 1] \to M$, let

$$L'(\gamma) := \int_0^1 F\left(x, f_x^{-1}\left(\frac{df \circ \gamma}{dt}(t)\right)\right) dt \tag{25}$$

and let $E(\gamma)$ be the Euclidean length of $f \circ \gamma$. We can take $B > 0$ such that, when a curve $\gamma$ is contained in a sufficiently small neighborhood of $x$, $L(\gamma) \geq BE(\gamma)$ (Busemann & Mayer, 1941).

Since $F$ is continuous and $S^{d-1}$ is compact, for any $\varepsilon > 0$, we can take $\eta > 0$ such that for any $y \in B_x(\eta) := \{p \in M | d(x, p) < \eta\}$ and $\mathbf{v} \in S^{d-1}$, $\left|F\left(x, f_x^{-1}(\mathbf{v})\right) - F\left(y, f_y^{-1}(\mathbf{v})\right)\right| < \varepsilon' := \varepsilon B/2$. Then, for any curve $\gamma$ contained in $B_x(\eta)$,

$$\begin{aligned}
|L(\gamma) - L'(\gamma)| &\leq \int_0^1 \left| F\left(\gamma(t), \frac{d\gamma}{dt}(t)\right) - F\left(x, f_x^{-1}\left(\frac{df \circ \gamma}{dt}(t)\right)\right) \right| dt \\
&\leq \varepsilon' \int_0^1 \left| \frac{df \circ \gamma}{dt}(t) \right| dt \\
&= \varepsilon' E(\gamma).
\end{aligned} \tag{26}$$

We can take $\delta \leq \eta$ such that for any $y, z \in B_x(\delta)$, there exists a minimizing geodesic $\gamma(y, z)$ from $y$ to $z$ contained in $B_x(\eta)$ (Bao et al., 2000). Note that $d(y, z) = L(\gamma(y, z))$. After retaking $\delta$ to be smaller if necessary, we can also assume that the inverse image $l(y, z)$ by $f$ of the segment from $f(y)$ to $f(z)$ is also contained in $B_x(\eta)$. Note that $|C(y, z) - L'(l(y, z))| \leq \varepsilon' \|f(y) - f(z)\|$.

Since $L'$ is a Minkowskian metric, $L'(l(y, z)) \leq L'(\gamma(y, z))$ (Busemann & Mayer, 1941). Therefore, for any $y, z \in B_x(\delta)$,

$$
\begin{aligned}
d(y, z) - 2\varepsilon' \|f(y) - f(z)\| &\leq L(l(y, z)) - 2\varepsilon' \|f(y) - f(z)\| \\
&\leq L'(l(y, z)) - \varepsilon' \|f(y) - f(z)\| \\
&\leq C(x, y) \\
&\leq L'(l(y, z)) + \varepsilon' \|f(y) - f(z)\| \\
&\leq L'(\gamma(y, z)) + \varepsilon' \|f(y) - f(z)\| \\
&\leq d(y, z) + \varepsilon' E(\gamma(y, z)) + \varepsilon' \|f(y) - f(z)\|.
\end{aligned}
\tag{27}
$$

Since $\|f(y) - f(z)\| \leq E(\gamma(y, z)) \leq B^{-1} d(y, z)$, the condition is satisfied.

## A.2 Proof of Proposition 2

To prove Proposition 2, we prove the following lemmas.

**Lemma 1.** *Let $(X, d)$ be a pseudo-quasi-metric space. Let $\pi_i : X \times X \to X$ and $V_i : X \times X \to \mathbb{R}$ be two series of functions indexed by $i \in \mathbb{N}$, such that for any $x, y \in X$,*

$$
\pi_i(x, y) \in \arg\min_z \left( V_i(x, z)^2 + V_i(z, y)^2 \right)
\tag{28}
$$

*and*

$$
V_{i+1}(x, y) = V_i(x, \pi_i(x, y)) + V_i(\pi_i(x, y), y).
\tag{29}
$$

*Let $\pi_\infty : X \times X \to X$ and $V_\infty : X \times X \to \mathbb{R}$ be two functions such that $d(\pi_\infty(x, y), \pi_i(x, y)) \to 0$ and $V_i(x, y) \to V_\infty(x, y)$ when $i \to \infty$ for any $x, y \in X$.*

1. *We assume that the series of functions $(V_i)_i$ is equicontinuous, i.e., for any $x, y \in X$ and $\varepsilon > 0$, there exists $\delta > 0$ such that for any $x', y' \in X$ and $i \in \mathbb{N}$*

$$
d(x, x') < \delta \wedge d(y, y') < \delta \implies |V_i(x, y) - V_i(x', y')| < \varepsilon.
\tag{30}
$$

*Then, for any $x, y \in X$,*

$$
\pi_\infty(x, y) \in \arg\min_z \left( V_\infty(x, z)^2 + V_\infty(z, y)^2 \right),
\tag{31}
$$

*and*

$$
V_\infty(x, y) = V_\infty(x, \pi_\infty(x, y)) + V_\infty(\pi_\infty(x, y), y).
\tag{32}
$$

2. *We assume that $(X, d)$ is weakly symmetric and $V_0(x, x) = 0$, $V_0(x, y) \geq 0$, and $V_0(x, y) = 0 \implies d(x, y) = 0$ for any $x, y \in X$. Then, for any $x \in X$,*

$$
d(x, \pi_\infty(x, x)) = d(\pi_\infty(x, x), x) = 0.
\tag{33}
$$

3. *For any $\varepsilon, \delta > 0$, if, for any $x, y \in X$,*

$$
V_0(x, y) < \delta \implies d(x, y) \leq (1 + \varepsilon) V_0(x, y),
\tag{34}
$$

*for any $x, y \in X$,*

$$
V_\infty(x, y) < \delta \implies d(x, y) \leq (1 + \varepsilon) V_\infty(x, y),
\tag{35}
$$

4. *We assume that $(X, d)$ has the midpoint property. For any $\varepsilon, \delta > 0$, if, for any $x, y \in X$,*

$$
d(x, y) < \delta \implies V_0(x, y) \leq (1 + \varepsilon) d(x, y),
\tag{36}
$$

*for any $x, y \in X$,*

$$
d(x, y) < \delta \implies V_\infty(x, y) \leq (1 + \varepsilon) d(x, y),
\tag{37}
$$

*Proof.* We prove 1. For any sequence $x_i$ and $y_i$ that converges $x_\infty$ and $y_\infty$ respectively,

$$|V_\infty(x_\infty, y_\infty) - V_i(x_i, y_i)| \le |V_\infty(x_\infty, y_\infty) - V_i(x_\infty, y_\infty)| + |V_i(x_\infty, y_\infty) - V_i(x_i, y_i)| \quad (38)$$

and the first term can be bound by convergence $V_i \to V_\infty$ at $(x_\infty, y_\infty)$ and the second term can be bound by equicontinuity of $(V_i)_i$. Thus, $\lim_{i\to\infty} V_i(x_i, y_i) = V_\infty(x_\infty, y_\infty)$. In particular,

$$\lim_{i\to\infty} V_i(x, \pi_i(x, y)) = V_\infty(x, \pi_\infty(x, y)), \quad (39)$$

$$\lim_{i\to\infty} V_i(\pi_i(x, y), y) = V_\infty(\pi_\infty(x, y), y). \quad (40)$$

By (28), for any $x, y, z \in X$,

$$V_i(x, \pi_i(x, y))^2 + V_i(\pi_i(x, y), y)^2 \le V_i(x, z)^2 + V_i(z, y)^2. \quad (41)$$

When taking a limit for $i \to \infty$,

$$V_\infty(x, \pi_\infty(x, y))^2 + V_\infty(\pi_\infty(x, y), y)^2 \le V_\infty(x, z)^2 + V_\infty(z, y)^2. \quad (42)$$

Thus, (31) follows. We can also show (32) by taking a limit of both sides in (29).

We prove 2. When $V_i(x, x) = 0$, since $V_i(x, \pi_i(x, x)) = V_i(\pi_i(x, x), x) = 0$ by (28), $V_{i+1}(x, x) = 0$ by (29). Thus, $V_i(x, x) = V_i(x, \pi_i(x, x)) = V_i(\pi_i(x, x), x) = 0$ for all $i$ by induction. Since we can also show $V_i(x, y) \ge 0$ for all $i$ by (29) and induction, $V_{i+1}(x, y) = 0$ indicates $V_i(x, \pi_i(x, y)) = V_i(\pi_i(x, y), y) = 0$ by (29). Using this, $V_i(x, y) = 0 \implies d(x, y) = 0$ for all $i$ is proven by induction. Therefore, $d(x, \pi_i(x, x)) = d(\pi_i(x, x), x) = 0$ for all $i$. By the assumption that $d$ is weakly symmetric, (33) is proven.

We prove 3. Since this assumption indicates $V_0(x, y) \ge 0$, by (29) and induction, $V_i(x, y) \ge 0$ for all $i$. Thus, $V_{i+1}(x, y) < \delta$ indicates $V_i(x, \pi_i(x, y)) < \delta$ and $V_i(\pi_i(x, y), y) < \delta$ by (29). We assume that $V_i(x, y) < \delta \implies d(x, y) \le (1 + \varepsilon)V_i(x, y)$ for all $x, y \in X$. Then, when $V_{i+1}(x, y) < \delta$, since $d(x, \pi_i(x, y)) \le (1 + \varepsilon)V_i(x, \pi_i(x, y))$ and $d(\pi_i(x, y), y) \le (1 + \varepsilon)V_i(\pi_i(x, y), y)$, $d(x, y) \le d(x, \pi_i(x, y)) + d(\pi_i(x, y), y) \le (1 + \varepsilon)V_{i+1}(x, y)$. Therefore, $V_i(x, y) < \delta \implies d(x, y) \le (1 + \varepsilon)V_i(x, y)$ for all $i$ by induction, which leads (35).

We prove 4. We assume that $d(x, y) < \delta \implies V_i(x, y) \le (1 + \varepsilon)d(x, y)$ for all $x, y \in X$. Let $x, y \in X$ be a pair such that $d(x, y) < \delta$. By assumption, there exists their midpoint $m$. Then,

$$\begin{aligned}
V_{i+1}(x, y)^2 &= (V_i(x, \pi_i(x, y)) + V_i(\pi_i(x, y), y))^2 \\
&\le 2V_i(x, \pi_i(x, y))^2 + 2V_i(\pi_i(x, y), y)^2 \\
&\le 2V_i(x, m)^2 + 2V_i(m, y)^2 \\
&\le 2(1 + \varepsilon)^2 \left( d(x, m)^2 + d(m, y)^2 \right) \\
&= (1 + \varepsilon)^2 d(x, y)^2,
\end{aligned} \quad (43)$$

where the first inequality comes from $(a + b)^2 \le 2a^2 + 2b^2$, the second comes from (29), and the third comes from the induction hypothesis and $d(x, m) = d(m, y) < \delta$. Thus, $d(x, y) < \delta \implies V_i(x, y) \le (1 + \varepsilon)d(x, y)$ for all $i$ by induction, which leads (37).

$\square$

**Lemma 2.** *Let $(X, d)$ be a pseudo-quasi-metric space with the midpoint property. Let $\pi : X \times X \to X$ and $V : X \times X \to \mathbb{R}$ be functions satisfying the following conditions.*

*1. For any $x, y \in X$,*

$$V(x, y) = V(x, \pi(x, y)) + V(\pi(x, y), y). \quad (44)$$

*2. For any $x, y \in X$,*

$$\pi(x, y) \in \arg\min_z \left( V(x, z)^2 + V(z, y)^2 \right), \quad (45)$$

*3. For any $x \in X$,*

$$d(x, \pi(x, x)) = d(\pi(x, x), x) = 0. \quad (46)$$

4. *For any $\alpha > 0$, there exists $\beta > 0$ such that for any $x, y, z \in X$,*

$$d(y, z) < \beta \implies d(\pi(x, y), \pi(x, z)) < \alpha \wedge d(\pi(y, x), \pi(z, x)) < \alpha. \tag{47}$$

*For a sufficient small $\varepsilon > 0$, if there exists $\delta > 0$ such that for any $x, y \in X$,*

$$d(x, y) < \delta \implies (1 - \varepsilon)d(x, y) \leq V(x, y) \leq (1 + \varepsilon)d(x, y), \tag{48}$$

*for any $x, y \in X$, $(1 - \varepsilon)d(x, y) \leq V(x, y) \leq (1 + \varepsilon)d(x, y)$.*

*In particular, if there exists such $\delta > 0$ for any $\varepsilon > 0$, $V = d$ and $\pi(x, y)$ is a midpoint between $x$ and $y$ for any $x, y \in X$.*

*Proof.* We first prove $|V(x, y)| \leq (1 + \varepsilon)d(x, y)$ for any $x, y \in X$. We prove $|V(x, y)| \leq (1 + \varepsilon)d(x, y)$ when $d(x, y) < 2^n \delta$ by induction for $n \in \mathbb{N}$. This is clear when $n = 0$. We assume it is true for $n$ and take $x, y \in X$ such that $d(x, y) < 2^{n+1}\delta$. Let $m$ be the midpoint between $x$ and $y$. Then, by a similar calculation with (43),

$$\begin{aligned}
V(x, y)^2 &= (V(x, \pi(x, y)) + V(\pi(x, y), y))^2 \\
&\leq 2V(x, \pi(x, y))^2 + 2V(\pi(x, y), y)^2 \\
&\leq 2V(x, m)^2 + 2V(m, y)^2 \\
&\leq 2(1 + \varepsilon)^2 \left( d(x, m)^2 + d(m, y)^2 \right) \\
&= (1 + \varepsilon)^2 d(x, y)^2
\end{aligned} \tag{49}$$

where the first equality comes from (44), the second inequality comes from (45), and the third inequality comes from the induction hypothesis and $d(x, m) = d(m, y) < 2^n \delta$. Thus, $|V(x, y)| \leq (1 + \varepsilon)d(x, y)$. Consequently, $|V(x, y)| \leq (1 + \varepsilon)d(x, y)$ for all $x, y \in X$.

Next, we prove $(1 - \varepsilon)d(x, y) \leq V(x, y)$ for any $x, y \in X$. By (46) and (47), we can take $\delta' \leq \delta$ such that

$$d(x, y) < \delta' \implies d(x, \pi(x, y)) \leq \delta \wedge d(\pi(x, y), y) \leq \delta. \tag{50}$$

By (47), we can take $0 < \eta < \delta'$ such that $d(\pi(x, y), \pi(x, z)) < \delta'/3$ and $d(\pi(y, x), \pi(z, x)) < \delta'/3$ when $d(y, z) < 2\eta$. We prove by induction for $n \in \mathbb{N}$ that

$$\begin{aligned}
d(x, y) < \delta' + n\eta \implies &(1 - \varepsilon)d(x, y) \leq V(x, y) \\
&\wedge (1 - \varepsilon)d(x, \pi(x, y)) \leq V(x, \pi(x, y)) \\
&\wedge (1 - \varepsilon)d(\pi(x, y), y) \leq V(\pi(x, y), y).
\end{aligned} \tag{51}$$

The case $n = 0$ follows from the conditions of $\delta$ and $\delta'$. We assume that it is true for $n$ and take $x, y$ such that $\delta' + n\eta \leq d(x, y) < \delta' + (n + 1)\eta$. By taking midpoints recursively, we can take $z \in X$ such that $d(x, z) + d(z, y) = d(x, y)$ and $\eta \leq d(z, y) < 2\eta$. Let $w := \pi(x, z)$, $a := V(x, w)$, $b := V(w, z)$, and $l := d(x, z)$. By (44) and the equality we previously proved, $a + b = V(x, z) \leq (1 + \varepsilon)l$. On the other hand, since $l < \delta' + n\eta$, by the induction hypothesis, $a + b = V(x, z) \geq (1 - \varepsilon)l$. Let $m$ be a midpoint between $x$ and $z$. Then, by (45) and the equality we previously proved,

$$a^2 + b^2 \leq V(x, m)^2 + V(m, z)^2 \leq (1 + \varepsilon)^2 \left( d(x, m)^2 + d(m, z)^2 \right) = \frac{(1 + \varepsilon)^2 l^2}{2}. \tag{52}$$

Thus,

$$\begin{aligned}
a &\leq \frac{1}{2} \left( a + b + |a - b| \right) \\
&= \frac{1}{2} \left( a + b + \sqrt{2(a^2 + b^2) - (a + b)^2} \right) \\
&\leq \frac{1}{2} \left( (1 + \varepsilon)l + \sqrt{(1 + \varepsilon)^2 l^2 - (1 - \varepsilon)^2 l^2} \right) \\
&= c_\varepsilon l,
\end{aligned} \tag{53}$$

where $c_\varepsilon := (1 + 3\varepsilon)/2$. By the induction hypothesis, $d(x, w) \leq (1 - \varepsilon)^{-1} a \leq (1 - \varepsilon)^{-1} c_\varepsilon l$. Since $\varepsilon$ is a sufficiently small value, we can assume that $(1 - \varepsilon)^{-1} c_\varepsilon < 2/3$. Let $p := \pi(x, y)$. Then,

$$d(x, p) \leq d(x, w) + d(w, p) < \frac{2}{3}(\delta' + n\eta) + \frac{1}{3}\delta' < \delta' + n\eta, \tag{54}$$

where the second inequality comes from $d(z, y) < 2\eta$ and the way $\eta$ is taken. Thus, by the induction hypothesis, $(1 - \varepsilon)d(x, p) \leq V(x, p)$. By the symmetrical argument, we can also prove $(1 - \varepsilon)d(p, y) \leq V(p, y)$. Then,

$$(1 - \varepsilon)d(x, y) \leq (1 - \varepsilon)(d(x, p) + d(p, y)) \leq V(x, p) + V(p, y) = V(x, y). \tag{55}$$

Therefore, $(1 - \varepsilon)d(x, y) \leq V(x, y)$ for any $x, y \in X$.

Thus, if the assumption holds for any $\varepsilon > 0$, $V = d$. Then, by (45) and the midpoint property, $\pi(x, y)$ is a midpoint between $x$ and $y$ for any $x, y \in X$. □

By the aforementioned lemmas, it is sufficient to show the following to prove Proposition 2.

1. The condition 4 in Lemma 2 holds for $\pi^*$, i.e., $\pi^*$ is uniformly continuous.

2. For any $\varepsilon > 0$, there exists $\delta > 0$ such that the assumptions of 3 and 4 in Lemma 1 are satisfied for $V_0^* = C$, i.e., the convergence $C(x, y)/d(x, y) \to 1$ is uniform.

These two properties follow from the following lemma, which is a generalization of a well-known fact for metric spaces to weakly symmetric pseudo-quasi-metric spaces.

**Lemma 3.** *Let $(X, d_X)$ be a compact pseudo-quasi-metric space and $(Y, d_Y)$ be a weakly symmetric pseudo-quasi-metric space. If a function $f : X \to Y$ is continuous, $f$ is uniformly continuous, i.e., for any $\varepsilon > 0$, there exists $\delta > 0$ such that $d_Y(f(x), f(y)) < \varepsilon$ for any $x, y \in X$ such that $d_X(x, y) < \delta$.*

*Proof.* We take $\varepsilon > 0$ arbitrarily. Since $Y$ is weakly symmetric, for any $x \in X$, we can take $\varepsilon(x) \leq \varepsilon/2$ such that for any $z \in Y$, $d_Y(f(x), z) < \varepsilon(x)$ indicates $d_Y(z, f(x)) < \varepsilon/2$. Since $f$ is continuous, we can take $\delta(x) > 0$ such that for any $y \in X$, $d_X(x, y) < 2\delta(x)$ indicates $d_Y(f(x), f(y)) < \varepsilon(x)$. Let $B(x) := \{y \in X | d_X(x, y) < \delta(x)\}$. Since $X$ is compact, we can take a finite $x_1, \ldots, x_N$ such that $B(x_1), \ldots, B(x_N)$ covers $X$. Let $\delta := \min_i \delta(x_i)$.

Then, for any $x, y \in X$, $d_X(x, y) < \delta$ indicates $d_Y(f(x), f(y)) < \varepsilon$. For we can take $x_i$ such that $d_X(x_i, x) < \delta(x)$. Since $d_Y(f(x_i), f(x)) < \varepsilon(x_i)$ follows from this, $d_Y(f(x), f(x_i)) < \varepsilon/2$. If $d_X(x, y) < \delta$, since $d_X(x_i, y) < 2\delta(x)$, $d_Y(f(x_i), f(y)) < \varepsilon(x_i) \leq \varepsilon/2$. Therefore, $d_Y(f(x), f(y)) < \varepsilon$. □

When $(X, d)$ is a pseudo-quasi-metric space, a pseudo-quasi-metric $d$ can be defined on $X \times X$ by $d((x_1, x_2), (y_1, y_2)) := d(x_1, y_1) + d(x_2, y_2)$ and the induced topology coincides with the topology as a direct product. Therefore, The condition 4 in Lemma 2 is a direct consequence of Lemma 3.

We define a function $r : X \times X \to \mathbb{R}$ as

$$r(x, y) = \begin{cases} \frac{C(x,y)}{d(x,y)} & d(x, y) \neq 0, \\ 1 & d(x, y) = 0. \end{cases} \tag{56}$$

Then, $r$ is continuous. When we take $x, y \in X$ and series $(x_i)_i$ and $(y_i)_i$ that converge $x$ and $y$, respectively, if $d(x, y) \neq 0$, since $d(x_i, y_i) \neq 0$ for a sufficient large $i$, $r(x_i, y_i) \to r(x, y)$ by continuity of $C$ and $d$. Otherwise, since $x_i \to x$ and $y_i \to x$, $r(x_i, y_i) \to 1$ by the assumption of $C$. By Lemma 3, $r$ is uniformly continuous. The existence of $\delta$ for the assumption of 4 in Lemma 1 follows from this.

To prove the existence of $\delta$ for the assumption of 3 in Lemma 1, it is sufficient to show that uniformly $d(x, y) \to 0$ when $C(x, y) \to 0$, i.e., for any $\varepsilon > 0$, there exists $\delta > 0$ such that $C(x, y) < \delta$ indicates $d(x, y) < \varepsilon$.

Take $\varepsilon > 0$ arbitrarily. For any $x \in X$, we can take $\delta(x) > 0$ such that for any $y \in X$, $C(x, y) < \delta(x)$ indicates $d(x, y) < \varepsilon/2$. Since $C$ is uniformly continuous by Lemma 3, we can take $\eta(x) > 0$

such that for any $y, z \in X$, $d(x, z) < \eta(x)$ indicates $|C(x, y) - C(z, y)| < \delta(x)/2$. Since $X$ is weakly symmetric, after retaking $\eta(x)$ to be smaller if necessary, we can also assume that, for any $z \in X$, $d(x, z) < \eta(x)$ indicates $d(z, x) < \varepsilon/2$. Since $X$ is compact, we can take a finite $x_1, \ldots, x_N$ such that for any $x \in X$, there exists $x_i$ such that $d(x_i, x) < \eta(x_i)$. Let $\delta := \min_i \delta(x_i)/2$.

We prove that $C(x, y) < \delta$ indicates $d(x, y) < \varepsilon$ for any $x, y \in X$. We can take $x_i$ such that $d(x_i, x) < \eta(x_i)$, which indicates $d(x, x_i) < \varepsilon/2$ and $|C(x_i, y) - C(x, y)| < \delta(x_i)/2$. When $C(x, y) < \delta \leq \delta(x_i)/2$, since $C(x_i, y) < \delta(x_i)$, $d(x_i, y) < \varepsilon/2$. Thus, $d(x, y) < \varepsilon$.

## B  IMPLEMENTATION DETAILS

Table 1: Values of Hyperparameters

|  | Ours | Seq | PG |
|---|---|---|---|
| Learning rate for § 5.3.1 | $3 \times 10^{-5}$ | $3 \times 10^{-3}$ | $5 \times 10^{-3}$ |
| Learning rate for § 5.3.2 and § 5.3.3 | $10^{-6}$ | $3 \times 10^{-3}$ | $5 \times 10^{-3}$ |
| Batch size | 256 | 128 | 300 |
| Number of epochs | 10 | 10 | 1 |
| Hidden layer sizes | [64, 64] | [64, 64] | [64, 64] |
| Activation function | ReLU | Tanh | Tanh |
| $\lambda$ for GAE | - | 0.95 | - |
| Clipping parameter | - | 0.2 | 0.2 |
| Entropy coefficient | - | 0 | 1 |
| VF coefficient | - | 0.5 | - |
| Max gradient norm | - | 0.5 | - |
| Base standard derivation | - | - | 0.05 |
| Number of samples per episode | - | - | 10 |
| Number of episodes per cycle | - | - | 30 |
| Total timesteps for § 5.3.1 | $2 \times 10^7$ | $2 \times 10^7$ | 20,613,600 |
| Total timesteps for § 5.3.2 | $8 \times 10^7$ | $8 \times 10^7$ | 82,591,200 |
| Total timesteps for § 5.3.3 | $2 \times 10^7$ | $2 \times 10^7$ | 20,032,200 |

Table 1 shows values of hyperparameters we used for our method and the baselines.

### B.1  OUR METHOD

At Line 2 of Alg. 1, we continue learning until the number of timesteps reach the defined value $T$. At Line 18, we gradually increase the depth of collecting data to the defined depth $D_{\max}$ while learning. The depth for the $c$-th call of the data collection procedure is $\lfloor c/c_d \rfloor$, where $c_d := \lfloor T/(2^{D_{\max}+1} - 1) \rfloor + 1$. At Line 4, if the size of the data returned from one running of the data collection procedure is larger than the batch size, it is called once. Otherwise, it is called until one mini-batch is filled. At Line 17, two points are sampled uniformly randomly from the coordination space of the manifold. We set the number of epochs $N_{\text{epochs}}$ to 10 and the batch size to 256.

The actor network outputs a Gaussian distribution with a diagonal coviariant matrix on the coordinate space. While data collecting or training, a prediction of the actor is sampled from the distribution with the mean and derivations output by the network. While evaluating, the means is returned as a prediction. We use a reparametrization trick to train the actor as in SAC (Haarnoja et al., 2018). If the prediction is outside valid coordinates, it is projected to the nearest one.

Both the actor and critic networks are multilayer perceptrons with two hidden layers of size 64. ReLU was selected as the activation function after trying ReLU and Tanh. The size of the output layer in the actor network is twice the dimension of the manifold, one half represents a mean and the other half represents logarithms of standard derivations. Adam (Kingma & Ba, 2014) is used as the optimizer. The learning rate is tuned to $3 \times 10^{-5}$ for § 5.3.1 and to $10^{-6}$ for § 5.3.2 and § 5.3.3. PyTorch (Paszke et al., 2019) is used for implementation. The environments are also imple-

mented using PyTorch. For the implementation of the robotics arm environment, we use PyTorch Kinematics (Zhong et al., 2023).

## B.2 SEQUENTIAL REINFORCEMENT LEARNING

Let $d$ be the dimension of the manifold. In the environment for reinforcement learning, the observation space is $2d$ dimensional and represents pairs of current and goal states. Whenever an episode starts, a start and goal point are sampled uniformly randomly from the coordination space of the manifold. The action space is $d$ dimensional. If $v$ is output by the agent for an observation $(f(p), f(g))$, the coordination of the next state $f(q)$ is calculated as

$$f(q) := f(p) + \frac{\varepsilon}{F(p, f_p^{-1}(v))} v. \tag{57}$$

If $f(q)$ is outside valid coordinates, it is projected to the nearest one. Since $q$ cannot be calculated when $v = 0$ exactly, in such a case, $q$ is set to $p$ and the agent receives reward $R := -100$ as a penalty. Otherwise, the reward is calculated by (14).

We use PPO implemented in stable baseline3 (Raffin et al., 2019), which uses PyTorch. The discount factor is set to one. The learning rate and batch size are tuned to $3 \times 10^{-3}$ and $128$ after searching in $\{3 \times 10^{-2}, 3 \times 10^{-3}, 3 \times 10^{-4}\}$ and $\{64, 128, 256\}$, respectively. Other hyperparameters are set to the default values in the library. Note that, the default architecture of networks for both the actor and critic are multilayer perceptrons with two hidden layers of size $64$, which are the same as those of the proposed method. Tanh was selected as the activation function after trying ReLU and Tanh.

## B.3 POLICY GRADIENT

We modified the author's implementation of subgoal-tree policy gradient (SGT-PG) available at `https://github.com/tomjur/SGT-PG`, which uses TensorFlow (Abadi et al., 2016). The hyperparameters values except the sizes of the hidden layers are the same as those in the original paper description.

The hidden layers of the policy networks are changed from the original to two of size $64$ to be the same as those of the other methods. Tanh is used as the activation function. The policy network outputs a Gaussian distribution with a diagonal covariant matrix on the coordinate space. The size of the output layers is $2d$, where $d$ is the dimension of the manifold. Let $m_1, \ldots, m_d, \sigma_1, \ldots, \sigma_d$ be the output for input $s, g$. The distribution mean is $(s + g)/2 + (m_1, \ldots, m_d)^\mathsf{T}$ and the standard derivation for the $i$-th coordinate is $\mathrm{Softplus}(\sigma_i) + (0.05 + \mathrm{Softplus}(c_i))\|s - g\|$, where $c_i$ is a learnable parameter. While predictions are sampled from distributions during the training of the policy, we take the means as predictions during the evaluation or training of other policies with higher indexes. If a prediction is outside valid coordinates, it is projected to the nearest one.

When we train $\pi_D$, we sample $30$ values of $(p_0, p_{2^D})$, start and goal points, uniformly randomly from the coordinate space of the manifold per training cycle. For each sampled pair, we sampled $10$ values of $p_{2^D-1}$ from the distribution outputted by $\pi_D$ and generated other waypoints deterministically by $\pi_{D-1}, \ldots, \pi_1$ for each sampled value. The average of the cost $c_\tau$ is used as the baseline $b(p_0, p_{2^D})$ in (16). The objective is that of PPO with an entropy coefficient of $1$ and clipping parameter of $0.2$. The optimizer is Adam with the learning rate set to $5 \times 10^{-3}$.

In the environments for § 5.3.1 and § 5.3.2, we train $\pi_1$ for $1000$ cycles and other $\pi_D$ for $538$ cycles; In the environment for § 5.3.3, we train each policy for $1077$ cycles. Note that the total timesteps for § 5.3.1 is $2 \times 300 \times 1000 + (4 + 8 + 16 + 32 + 64) \times 300 \times 538 = 20613600 \approx 2 \times 10^7$, that for § 5.3.2 is $2 \times 300 \times 1000 + (4 + 8 + 16 + 32 + 64 + 128 + 256) \times 300 \times 538 = 82591200 \approx 8 \times 10^7$, and that for § 5.3.3 is $(2 + 4 + 8 + 16 + 32) * 300 * 1077 = 20032200$.

## C REMARKS ON THE SETTING

**Remark 5.** Instead of (4), if we set as

$$C(x, y) := \int_0^1 F\left(f^{-1}((1-t)f(x) + tf(y)), df_{f^{-1}((1-t)f(x)+tf(y))}^{-1}(f(y) - f(x))\right) dt, \tag{58}$$

which is the length of the curve connecting points as a segment in the coordinate space, $C$ gives upper bounds of distances. Thus, we can use the setting of Jurgenson et al. (2020). However, the integration (58) is not always computable efficiently.

**Remark 6.** Instead of (4), we may define $C$ as

$$C(x, y) := \frac{1}{2} \left( F\left(x, df_x^{-1}(f(y) - f(x))\right) + F\left(y, df_y^{-1}(f(y) - f(x))\right) \right).$$ (59)

Since it is not biased toward the $x$ side, it might be more appropriate for approximating distances. The main reason we do not adopt this definition in this paper is the difficulty in using it in sequential reinforcement learning, one of the baseline methods in our experiments.

**Remark 7.** For pseudo-Finsler manifolds, which are similar to Finsler manifolds but do not necessarily satisfy the condition of positive definiteness, the distance function $d$ can be defined and is a weakly symmetric quasi-metric (Javaloyes & Sánchez, 2011). However, we do not expect that Proposition 1 holds for pseudo-Finsler manifolds generally.

## D  REMARKS ON UNIDIRECTIONAL CAR-LIKE CONSTRAINTS

While the metric we have defined in § 5.3.2 is not strictly a Finsler metric, we define distances $d$ and the function $C$ using $F$ in the same way as Finsler metrics. To make it a Finsler metric, we have to replace the definition of $\xi$ with their smooth approximations. Note that $F((p, \theta), -)^2$ is convex and its Hessian matrix is positive definite where it is smooth.

In terms of Fukuoka & Setti (2021), $F$ is a $C^0$-Finsler structure, which can be approximated by Finsler structures.

**Remark 8.** While the vehicle model in Rösmann et al. (2017) is bidirectional, i.e., it can move backward, our model is unidirectional, i.e., it can only move forward. Unidirectionality seems to be essential for modeling by a Finsler metric. If we replace (24) with $\xi := \max\{r_{\min}|v_\theta| - |v_x \cos(\theta) + v_y \sin(\theta)|, 0\}$, $F$ cannot be approximated by Finsler structures since it does not satisfy the subadditivity ($F(x, v + w) \leq F(x, v) + F(x, w)$).

## E  MORE ON THE ROBOTIC ARM MOTION PLANNING TASK

### E.1  DEFINITION OF ENVIRONMENT

Let $x, y, z$ be the coordinate system of the workspace. Suppose there is an obstacle parallel to the $z$-axis at $x = o_x, y = o_y$.

Let $M$ be the configuration space for Franka Panda robotic arm. Each component of a state in $M$ represents an angle of the corresponding joint. Franka Panda robotic arm has 10 links including `panda_link0` and `panda_hand`. For $s \in M$, let $p_i(s)$ be the position of the $i$-th link, where $0 \leq i \leq 9$. We define $F$ as

$$F(s, v) := \left(1 + \sum_{i=0}^{9} \left((p_i(s)_x - o_x)^2 + (p_i(s)_y - o_y)^2\right)^{-1/2}\right) \|v\|.$$ (60)

That is, if any one of the links gets closer to the obstacle, the value of the metric increases. Thus, geodesics can be expected to avoid the obstacle.

We set $(o_x, o_y) := (0.4, 0.0)$ and $\varepsilon = 20$.

### E.2  EXAMPLE OF GENERATED MOTION

The middle row in Fig. 4 shows an example of a motion generated by the policy learned by our proposed method. For comparison, the motion generated by linear interpolation in the configuration space and the ground truth of the geodesic with the same endpoints are also drawn. We used Robotics Toolbox for Python (Corke & Haviland, 2021) for visualization.

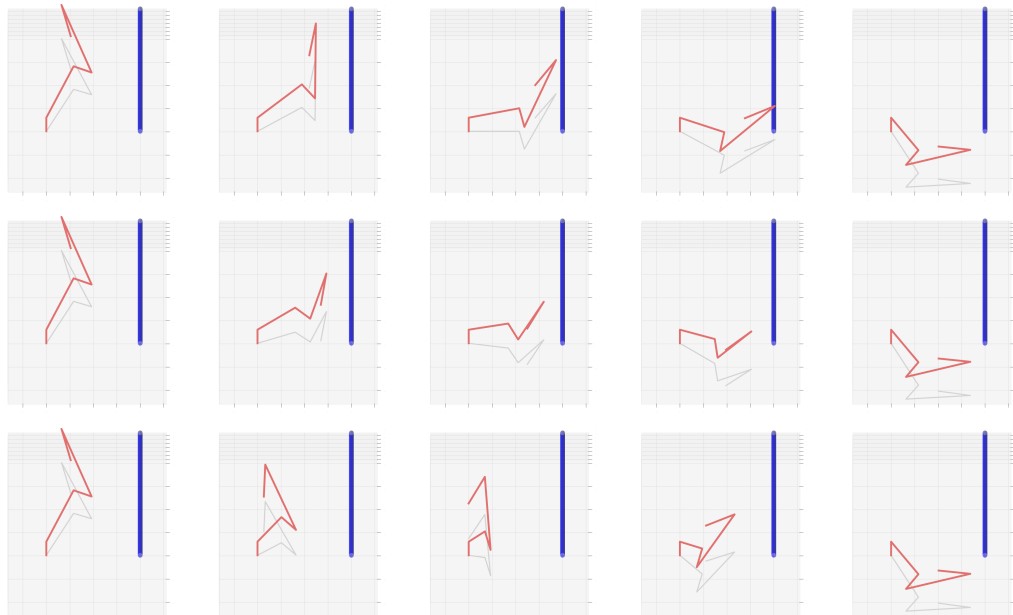

Figure 4: Examples of motions. The first row shows linear interpolation. The second row shows a generated motion by learned policy. The third row shows the true geodesic. The pole is an obstacle.

In this example, the motion by mere linear interpolation causes the arm to hit the obstacle. The motion generated by the learned policy succeeds in avoiding the obstacle. This shows that the proposed method is effective for collision-free motion planning of a robotic arm.

However, depending on endpoints, non-joint parts of the arm sometimes collide with the obstacle in generated motions. This is due to the fact that the metric (60) only considers distances between the joints and the obstacle, and needs to be improved to also consider parts between joints.

