# OpenReview forum: "Generation of Geodesics with Actor-Critic Reinforcement Learning to Predict Midpoints"
_ICLR.cc/2024/Conference — Submitted to ICLR 2024_

### Official Review · Reviewer_8vLp · 2023-10-19

**Soundness:** 3 good
**Presentation:** 2 fair
**Contribution:** 2 fair
**Rating:** 3
**Confidence:** 3

**Summary:**

In this paper, the authors study the problem of finding geodesics in general manifolds via reinforcement learning. The main idea is to divide the discovery task into smaller ones by predicting midpoints recursively. An actor-critic algorithm learns a policy to generate midpoints. Two empirical evaluations are provided to demonstrate the efficacy of the proposed algorithm.

**Strengths:**

1. Geodesics generation with reinforcement learning is a relatively under-explored research area. This work contributes by studying an actor-critic formulation and shows its effectiveness.

2. A few design choices are explored, such as different variants of the actor loss. These results help illustrate some properties of the proposed algorithm.

**Weaknesses:**

1. The empirical evaluation environments are relatively artificial. I would expect some more practical tasks such as robotic motion planning to be more effective in demonstrating the significance of the contribution.

2. The current baselines are all RL based. I think some classical motion planning algorithms should be included too, such as RRT (RRT*) and A* search.

3. This is more of a clarification of the problem setting. It seems that the end goal of the learned policy is not necessarily finding the shortest path. The success criterion is stated as “all values of C(4) for two consecutive points are not greater than $\epsilon$”, which does not imply that a path is the shortest. Is this correct? If so, this should be stated more clearly.

**Questions:**

1. Please provide some motivations for the definition of $C$ (Equation (4)). Also please explain what $df_x$ is in this definition.

2. Why is Equation (5) hard to compute efficiently?

3. How does one decide the depth parameter $D$ on Line 18 of Algorithm 1?

4. In Equation (11), should the right-hand side be $d(x, y)$, the true distance rather than the local approximation? Either way, Equation (11) could use a more expanded explanation.

5. In Proposition 2, what is $V_i$?

6. In the Sequential Reinforcement Learning (Seq) baseline, why is the reward function (Equation (16)) scaled by $\epsilon$? How does this decision affect the learning?

---

> ### Author Response · Authors · 2023-11-13
>
> Thank you for the review!
>
> > This is more of a clarification of the problem setting. It seems that the end goal of the learned policy is not necessarily finding the shortest path. The success criterion is stated as “all values of C(4) for two consecutive points are not greater than $\epsilon$”, which does not imply that a path is the shortest. Is this correct? If so, this should be stated more clearly.
>
> Indeed, judged as a success does not mean being the shortest, but success rates should increase if shorter paths are chosen for all pairs. The reason we do not use the lengths of paths for evaluation is that, since the metric can only be calculated locally, lengths cannot be calculated when the success condition is satisfied. Moreover, it is needed to compare fairly with the sequential baseline method, which has a different generation method. For these reasons, we chose this evaluation scheme.
>
> > Please provide some motivations for the definition of $C$
>  (Equation (4)). Also please explain $df_x$ what
>  is in this definition.
>
> Sorry for the missing explanation. $df_x$ is the differential (pushfoward) of $f$ at $x$.
>
> Since the Finsler distance is defined by (1) and the geodesics can be approximated by the segment in the coordinate space,
> we approximate the distance between the closed points by (5). Furthermore, since $F$ is continuous and the points are closed, this integral can be approximated by the value at $t=0$. This is (4).
>
> > Why is Equation (5) hard to compute efficiently?
>
> Of course, there are various methods of numerical calculation of integrals, but they take more time than calculating a function once
> and it is difficult to know in advance the required accuracy.
>
> > In Equation (11), should the right-hand side be $d(x,y)$, the true distance rather than the local approximation? Either way, Equation (11) could use a more expanded explanation.
>
> Unlike the proposed method, different policies and value functions are considered here for different depths.
> $V_i$ represents the sums of the values of $C$ after the policies $\pi_{i-1}, \ldots ,\pi_0$ is applied.
> It is therefore correct that the right-hand side of (11) is $C$.
>
> > In Proposition 2, what is $V_i$ ?
>
> This is a typo of $V_i^*$. Thank you for pointing this out.
>
> > In the Sequential Reinforcement Learning (Seq) baseline, why is the reward function (Equation (16)) scaled by $\epsilon$ ? How does this decision affect the learning?
>
> This motivates agents to reach goals in as few steps as possible.

---

> ### Author Response · Authors · 2023-11-22
> **About baselines**
>
> The first concern that the experimental environments were artificial could have been answered by conducting an additional experiment.
>
> One of the reasons we focused the baselines on reinforcement learning is that it is difficult to make a fair comparison with searching methods.
> While searching methods can find good solutions if enough time is spent, prediction by trained policy uses very little time.
> Therefore, we decided to limit ourselves to RL methods for situations where we want to generate paths or motions in a short time.

---

> > ### Comment · Reviewer_8vLp · 2023-11-22
> > **Thanks for the response**
> >
> > I want to thank the authors for their responses. If the argument about classical planning methods such as RRT is the computation time, the authors should provide relevant evaluations to back it up. In particular, there are accelerated versions of those methods which can be made very fast as well. I will maintain my score.

---

> > > ### Author Response · Authors · 2023-11-23
> > >
> > > On computational time of searching methods, while there is no time for additional experiments, the following considerations can be made.
> > >
> > > The size of space to search for the success condition of our experiments is roughly $\Theta(\epsilon^{-d})$, where $d$ is the dimension of the space.
> > > Although the search can be speeded up if an approximate distance is known a priori, we assume that such information is not available.
> > > On the other hand, the time taken for generation by our method is $O(\log\epsilon^{-1})$ if parallelized.
> > > Thus, while searching methods are not scalable for $\epsilon$ value, whereas our method is.

---

### Official Review · Reviewer_4ppY · 2023-10-25

**Soundness:** 3 good
**Presentation:** 3 good
**Contribution:** 2 fair
**Rating:** 5
**Confidence:** 1

**Summary:**

This work focuses on path planning to generate geodesics in some manifold. It extends sub-goal tree framework (Jurgenson et al., 2020) to generate midpoints (equal distances to two given points), instead of any intermediate points. They train an actor to predict the midpoints, and a critic to predict the distance of two given points (s,g). It is also shown to converge to a unique optimal solution,  where the distance is given by some continuous approximation. The method is evaluated on two toy tasks to showcase its effectiveness over RL and previous planning approach.

**Strengths:**

The overall writing is rigorous, principled and looks solid work. But I am not sure of its significance.

**Weaknesses:**

Perhaps the motivation of this work can be better written. As the authors pointed out in their experiments, generating geodesic (path planning) can be simply tackled by RL by specifying a reward function related to the difference in distance. But it may have instability or other issue compared to path planning approaches.

Could you give some explanation why Car-like task favors your approach, while Matsumoto task not?

The experiment scope is a bit narrow as only two toy tasks are evaluated.

Minor: The description of methods in the experiments can be more complete – add a line of “ours” using Eq. 8 before “the following variants of our methods”.  The name “sequential RL” is a bit confusing as RL is sequential in nature. Perhaps “vanilla RL” or just “RL”, because your approach uses a non-conventional actor loss.

**Questions:**

I’m not familiar with path planning and differential manifold, so some of these comments are my educational guess.

---- Post-rebuttal

After reading the authors' response and other reviews, I think this work still requires more empirical evaluation on their approach. Thus, I lower my rating.

---

> ### Author Response · Authors · 2023-11-13
>
> Thank you for the review!
>
> > Perhaps the motivation of this work can be better written. As the authors pointed out in their experiments, generating geodesic (path planning) can be simply tackled by RL by specifying a reward function related to the difference in distance. But it may have instability or other issue compared to path planning approaches.
>
> We are grateful for your useful comments. Indeed, we felt that sequential RL is unstable for both random seed and hyperparameters compared to the proposed methods. It may be due to the fact that sequential RL have long horizons compared to the proposed method.
>
> > Could you give some explanation why Car-like task favors your approach, while Matsumoto task not?
>
> I think that sequential RL is not good at CarLike rather than our method is good at CarLike.
> Since sequential RL uses ad-hoc reward function defined in (16), there is no guarantee that it works well.
> Perhaps the defined reward is far from the actual distance in CarLike.
>
> > Minor: The description of methods in the experiments can be more complete – add a line of “ours” using Eq. 8 before “the following variants of our methods”. The name “sequential RL” is a bit confusing as RL is sequential in nature. Perhaps “vanilla RL” or just “RL”, because your approach uses a non-conventional actor loss.
>
> We will add a little explanation of what "Ours" means.
>
> The name "sequential" may be somewhat inappropriate, but I wouldn't dare change it because previous work by Jurgenson et al. have used this word.

---

### Official Review · Reviewer_ZvgN · 2023-11-03

**Soundness:** 3 good
**Presentation:** 3 good
**Contribution:** 3 good
**Rating:** 6
**Confidence:** 3

**Summary:**

The paper proposes a modification of the sub-goal tree framework to use midpoints instead of arbitrary intermediate points and actor-critic instead of policy gradient for goal-conditioned reinforcement learning problems. With the two changes, the proposed method is able to generate equally divided waypoints and with better sample efficiency on deep trees. Theoretical proofs are given for the convergence of the proposed method. The proposed method shows comparable performance to baselines on several tasks with advantage of generating equally divided waypoints.

**Strengths:**

The paper is well-written and the method is well-motivated. The effectiveness of the proposed method is supported both theoretically and empirically. The generated waypoints with equal distances would be more useful than that of the previous method.

**Weaknesses:**

The novelty of the paper is not prominent compared to its base methods.
The experimental setting is a bit simplified. In section 6, the authors propose a penalty term to be added to deal with obstacles. Wondering how easy is it to generalize the proposed method to environments with obstacles.
The experiment results do not show clear performance improvements of the proposed method.

**Questions:**

Can we add some more explanation and justification on why the midpoint is not just a trivial extension of the existing method using arbitrary waypoints?

Can we add more analysis on how the proposed method could be generalized to environments with obstacles?

In Figures 2 and 3, the proposed method does not show clear improvements compared to baselines. Is this expected? Can we add more explanations?

---

> ### Author Response · Authors · 2023-11-13
> **Answers to questions**
>
> Thanks for the review!
>
> > Can we add some more explanation and justification on why the midpoint is not just a trivial extension of the existing method using arbitrary waypoints?
>
> We think that he most non-trivial point for midpoint generation compared to the sub-goal tree generation is the theoretical soundness proved in the subsection 4.3. While the existing method using arbitrary waypoints can perform badly in our setting as in the situation of Remark 6, our midpoint generation method avoids this.
>
> > Can we add more analysis on how the proposed method could be generalized to environments with obstacles?
>
> Please see the papers cited in our response to Reviewer etxs.
>
> > In Figures 2 and 3, the proposed method does not show clear improvements compared to baselines. Is this expected? Can we add more explanations?
>
> In CarLike environments, the proposed method clearly outperformed other methods except the intermediate point variant of the proposed method.
> In Matsumoto environments, the proposed method outperformed other methods except the sequential RL baseline.
> Indeed, the superiority of the proposed method over the intermediate point variant may not be so obvious.
> However, it is noted that the intermediate point variant can perform badly in Remark 6.
> Thus, our method has been shown to be theoretically or experimentally superior to all other methods.

---

### Official Review · Reviewer_etxs · 2023-11-06

**Soundness:** 3 good
**Presentation:** 3 good
**Contribution:** 2 fair
**Rating:** 5
**Confidence:** 2

**Summary:**

This paper presents a novel reinforcement learning framework, termed as 'midpoint tree', designed to recursively generate geodesics for path planning. The approach introduces an actor-critic learning method tailored to predict midpoint waypoints, facilitating the construction of paths in complex environments. The paper details both the theoretical underpinnings and the practical implications of the method, demonstrating its application to two distinct metrics, the Matsumoto metric and a car-like metric, and discussing its potential in fields such as image processing and physical systems modeling.

**Strengths:**

The paper presents a distinct approach to generating geodesics in reinforcement learning environments via a "midpoint tree" algorithm. The theoretical underpinnings are robust, complemented by a thorough experimental evaluation. The articulation is commendable, with the authors elucidating complex ideas succinctly. This work's originality and potential applicability are clear, indicating its prospective value in advancing research within reinforcement learning and robotics.

**Weaknesses:**

The paper lacks a broader range of examples to demonstrate the applicability of the method to more common robotic tasks like locomotion and manipulation planning. The experimental results, while encouraging, do not showcase a significant advantage over existing methods, raising questions about the practical benefits of the proposed approach. It requires certain assumptions that may not be present in typical robotic environments, such as the need for global coordinate systems and uniform sampling. The method might not be readily applicable to more complicated, dynamic environments.

More concretely:
- The algorithm requires additional assumptions that may not be readily available or applicable in common robotic tasks, such as locomotion and manipulation planning. These assumptions include the need for global coordinate systems, obstacle-free environments, and environment-specific policy learning. The method's effectiveness is contingent on these conditions, which are not always present in more complex or dynamically changing real-world scenarios. Additionally, the challenge of generating globally optimal paths and dealing with the complexity of Finsler geodesics further limits its applicability to standard reinforcement learning tasks.

- In the original wording, the paper mentions that the method "only works well locally since we assume that manifolds have global coordinate systems and the continuous midpoint property may be satisfied only locally. For the generation of globally minimizing geodesics, we may have to divide manifolds, train policies for each local region and connect locally generated geodesics." It also states that "the policy has to be learned for each environment. By modifying our method so that the actor and critic input information on environments, it may be possible to learn a policy applicable to different environments." These statements highlight the limitations regarding the need for specific geometric and topological assumptions that may not hold in typical RL tasks in robotics.

A line of work on quasimetric distance for goal-conditioned RL seems related, which could provide important context and benchmarking. I'd be curious whether the proposed approach is related to them.
- Tongzhou Wang et al., Optimal Goal-Reaching Reinforcement Learning via Quasimetric Learning, ICML 2023
- Tongzhou Wang et al., On the Learning and Learnability of Quasimetrics, ICLR 2022

**Questions:**

- Can the authors provide additional examples where their method might be applicable, specifically within the realm of robotics tasks like locomotion and manipulation?
- How does the proposed approach compare in terms of benefits and applicability to other realistic tasks, beyond what has been demonstrated in the paper?
- Could the authors discuss the relationship and distinctions between their work and recent research on quasimetric learning for goal-conditioned RL?

---

> ### Author Response · Authors · 2023-11-13
> **Answers to questions**
>
> Thanks for the review!
>
> > Can the authors provide additional examples where their method might be applicable, specifically within the realm of robotics tasks like locomotion and manipulation?
>
> Some robotics tasks including manipulation are formulated by Riemannian geometry in the context of Riemannian Motion Optimization.
> See the following papers:
>
> Ratliff, Nathan, Marc Toussaint, and Stefan Schaal. "Understanding the geometry of workspace obstacles in motion optimization." 2015 IEEE International Conference on Robotics and Automation (ICRA). IEEE, 2015.
>
> Mainprice, Jim, Nathan Ratliff, and Stefan Schaal. "Warping the workspace geometry with electric potentials for motion optimization of manipulation tasks." 2016 IEEE/RSJ International Conference on Intelligent Robots and Systems (IROS). IEEE, 2016.
>
> Therefore, our method is potentially appliable to these tasks.
> Especially, since the technique to warp metrics for avoiding obstacles is already studied, our method can be appliable to environments with obstacles.
>
> The sentence "only works well locally since we assume that manifolds have global coordinate systems and the continuous midpoint property may be satisfied only locally" we wrote is maybe misleading. We assumed the continuous midpoint property for the theoretical result, it seems not to be practically necessary. Indeed, even environments we used for experiments seem not to have the property.
> Therefore, this assumption is a limitation of our theoretical contribution and not of our method.
>
> > How does the proposed approach compare in terms of benefits and applicability to other realistic tasks, beyond what has been demonstrated in the paper?
>
> In the context of aforementioned Riemannian motion optimization, the practical advantage of RL methods compared to the optimization method is the computational cost. While optimization is computationally costly, predictions by learned agents are fast. The advantage of our method over sequential RL method is that there is no need to reward engineering. In addition, the fact that the proposed method has a very short "horizon" compared to sequential RL may stabilize learning.
>
> > Could the authors discuss the relationship and distinctions between their work and recent research on quasimetric learning for goal-conditioned RL?
>
> Thank you for sharing related works!
> The main difference between our baseline sequential RL and the quasimetric learning method is architectures of value function approximation models.
> Since we do not focus on architectures of critics and used simple MLPs for both the proposed method and the baseline methods, the direction of our research and the direction of this research are orthogonal. The embedding method for quasimetric learning can be used for the critic in our method.

---

> > ### Author Response · Authors · 2023-11-22
> >
> > > How does the proposed approach compare in terms of benefits and applicability to other realistic tasks, beyond what has been demonstrated in the paper?
> >
> > I would like to add a few words on this point.
> >
> > Compared to sequential generation methods, generation by our method has the advantage that it can be easily parallelized.
> > Therefore, learning was faster, even for tasks with complex metrics.

---

### Author Response · Authors · 2023-11-22
**Additional Experiment**

Since it is pointed out that our experiments are not enough, we conducted an additional experiment.

This experiment involves motion planning for 7 DoF robotic arm in an environment with an obstacle.
See 5.3.3 and Appendix E in the revised article.

Results show that our method is also more effective for this task than the baselines.

In the previous resubmission, explanations were added where questions were asked.

To fit the constraint of the pages, we turned some unimportant remarks to appendices.
Some expressions were also changed to make the text shorter.

---

### Meta-Review · Area_Chair_ouwU · 2023-12-09

**Metareview:**

This paper introduces a novel RL method for path planning called midpoint tree. It modifies the existing sub-goal tree framework by focusing on generating midpoints rather than arbitrary intermediate points. An actor-critic learning approach is employed, with the actor predicting midpoints and the critic estimating distances. This method enhances sample efficiency and ensures equally divided waypoints. Authors provide theoretical convergence proofs under various assumptions, and the framework's effectiveness is demonstrated through two distinct metrics -- the Matsumoto metric and a car-like metric. This approach is shown to be comparable to existing methods in performance.

The reviewers of the paper raised several key concerns, primarily centered on the practical applicability and experimental scope of the proposed reinforcement learning framework for path planning. They highlighted limitations such as the method's reliance on specific assumptions around global coordinate systems and obstacle-free environments. The narrow focus of the experiments was also a point of contention, with suggestions to include more varied and practical tasks, and to compare the method with both RL-based and classical motion planning algorithms like RRT and A* search.

The authors did provide clarifications on their success criteria and algorithmic choices. They also noted that their method has better computational efficiency and scalability in comparison with classical search methods. Despite these clarifications, the reviewers remained unconvinced, reiterating the need for more comprehensive empirical evaluations and suggesting that comparisons with accelerated versions of classical planning methods would be beneficial. I agree with the reviewer concerns and the paper would be stronger and better grounded after taking this feedback into consideration.

**Justification For Why Not Higher Score:**

N/A

**Justification For Why Not Lower Score:**

N/A

---

### Decision · Program_Chairs · 2024-01-16

Reject